*Manuscript for*

# Source-Dependent Optical Properties and Molecular Characteristics of Atmospheric Brown Carbon

**Authors**: Jinghao Zhai[1,3], Yin Zhang[1,2], Pengfei Liu[4], Yujie Zhang[1,2], Antai Zhang[1,2], Yaling
Zeng[1,2], Baohua Cai[1,2], Jingyi Zhang[1,2], Chunbo Xing[1,2], Honglong Yang[5], Xiaofei Wang[3],
Jianhuai Ye[1,2], Chen Wang[1,2], Tzung-May Fu[1,2], Lei Zhu[1,2], Huizhong Shen[1,2] , Shu Tao[1,2], Xin
Yang[1,2]*
*[1]Shenzhen Key Laboratory of Precision Measurement and Early Warning Technology for Urban*
*Environmental Health Risks, School of Environmental Science and Engineering, Southern*
*University of Science and Technology, Shenzhen 518055, China*
*[2]Guangdong Provincial Observation and Research Station for Coastal Atmosphere and Climate*
*of the Greater Bay Area, Shenzhen 518055, China*
*[3]Shanghai Key Laboratory of Atmospheric Particle Pollution and Prevention (LAP[3]), Department*
*of Environmental Science and Engineering, Fudan University, Shanghai 200438, China*
*[4]School of Earth and Atmospheric Sciences, Georgia Institute of Technology, Atlanta, GA 30332,*
*USA*
*[5]Shenzhen National Climate Observatory, Meteorological Bureau of Shenzhen Municipality,*
*Shenzhen 518040, China*
*To whom correspondence should be addressed.
Correspondence to: Xin Yang
Email: yangx@sustech.edu.cn

**ABSTRACT:** Atmospheric brown carbon (BrC) can significantly affect Earth's radiation budget by its wavelength-dependent absorption in the ultraviolet (UV)-visible range. BrC consists of a wide variety of organics with different optical properties, making accurate climate modeling essential for understanding its radiative impact. Here, we conducted a field campaign during the summer in Shenzhen, China, to investigate the optical properties and molecular characteristics of BrC from diverse particle sources using both online and offline measurements. BrC mass concentrations were determined either based on thermally desorbed organic carbon or water-soluble organic carbon, and the corresponding mass absorption cross-sections (MAC) were calculated accordingly. Different sources of BrC, including those from secondary production associated with ozone pollution, urban transportation, and biomass burning, were identified through meteorological data and particle chemical compositions. The results show that the MAC of BrC varied across sources, with BrC from biomass combustion exhibiting the highest MAC at 370 nm ($3.42 \pm 0.41$ m$^2$/g) and secondary BrC associated with ozone pollution showing the lowest ($1.25 \pm 0.56$ m$^2$/g). Nevertheless, secondary BrC exhibited the highest absorption Ångström exponent (AAE) while the BrC from biomass burning had the lowest AAE. Molecular analysis revealed that species in the CHON family from biomass burning demonstrated the strongest light absorption. Our results provide valuable insights for quantifying the source-specific optical properties of BrC, enhancing the accuracy of climate models.

**Graphical abstract**

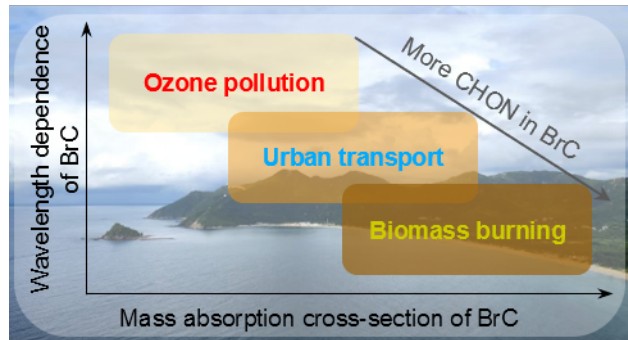

## 1 INTRODUCTION

Atmospheric light-absorbing organic aerosols, known as brown carbon (BrC), are important contributors to the global radiation absorption of atmospheric aerosols, alongside black carbon (BC). The absorption properties of BrC are wavelength-dependent, with relatively weak absorption in the mid- and long-visible wavelengths and a pronounced increase in absorption toward the short-visible and near-ultraviolet (UV) wavelengths (Sun et al., 2007;Laskin et al., 2015). Atmospheric BrC is primarily generated from the combustion of biomass and biofuels, as these processes typically occur under relatively low-temperature, fuel-rich conditions, which promote the formation of organics (Saleh et al., 2014;Chen and Bond, 2010). Additionally, secondary reactions in the atmosphere also play a significant role in the production of BrC (Laskin et al., 2015;Moise et al., 2015). It has been observed that BrC can be formed in secondary organic aerosols (SOA) through the nitration of volatile organic compound (VOC) precursors (Zhong and Jang, 2011;Lambe et al., 2013;Updyke et al., 2012;Haynes et al., 2019), aqueous-phase reactions of ammonia or amino acids with carbonyl-containing SOA (Updyke et al., 2012;Flores et al., 2014;Zarzana et al., 2012), and bond-forming reactions among SOA constituents that generate dimers and larger oligomers (Shapiro et al., 2009;Bones et al., 2010;Chang and Thompson, 2010). Unlike BC, which exhibits relatively uniform physicochemical properties, BrC comprises a broad spectrum of light-absorbing organic species, resulting in large variability in its optical properties (Updyke et al., 2012;Saleh et al., 2018). To accurately assess the radiative impacts of BrC, its diverse properties must be effectively represented in climate models.

Aerosol light absorption can be quantified using the mass absorption cross-section (MAC), a key parameter that links radiative transfer to aerosol mass in climate models (Bond and Bergstrom, 2006). MAC can be calculated from measurements of aerosol light absorption coefficient and mass concentration. The absorption Ångström exponent (AAE) describes the wavelength dependence of aerosol light absorption. For BC aerosols, AAE values are typically close to 1 (Bond and Bergstrom, 2006). In contrast, BrC shows substantial variability in wavelength dependence, with AAE values ranging from 2 to as high as 11 (Laskin et al., 2015). The optical properties of BrC

are highly source-dependent (Saleh et al., 2014;Kumar et al., 2018). Moreover, BrC absorption
evolves dynamically during atmospheric aging through processes like photobleaching or photo-
enhancement, leading to uncertainties in the quantification of the radiative effects of atmospheric
aerosols (Wong et al., 2017;Sumlin et al., 2017;Li et al., 2020).
The measurement of the optical properties of BrC is crucial for accurately determining its role
in global radiation balance. In filter-based offline analysis, the optical properties of a bulk film can
be measured using an ultraviolet-visible (UV-vis) spectrometer (Zhong and Jang, 2011). While the
varying solubility of BrC components in different solvents could introduce uncertainties in offline
analyses (Shetty et al., 2019), solvent-induced chemical artifacts, particularly those associated with
methanol extraction, have also been shown to significantly alter the optical properties of BrC
(Kumar et al., 2018;Saleh et al., 2014). Therefore, online MAC measurements provide a more
consistent and reliable benchmark, and integration with carefully selected offline extractions can
offer a more comprehensive understanding of the relative abundance of BrC classes (Chen et al.,
2022b). Nevertheless, the optical properties of BrC retrieved from online measurements can be
subject to biases due to the limitations of the techniques employed. For example, transmission
measurements through aerosol-laden filters have been used to quantify aerosol absorption
properties (Petzold et al., 2005;Bond and Bergstrom, 2006). These approaches usually assume that
aerosol particles retain their morphology upon adhering to the filters, potentially leading to
uncertainties in the interpretation of filter absorption data (Subramanian et al., 2007). Various
online approaches have been developed to directly measure the absorption, scattering, and
extinction coefficients of aerosols, either independently or in combination. Cavity-based
techniques offer highly sensitive and accurate measurements of the overall extinction coefficient
(Riziq et al., 2007;Massoli et al., 2010). An integrating nephelometer enables the independent
measurement of the scattering coefficient (Anderson and Ogren, 1998;Bond et al., 2009).
Photoacoustic instruments are widely recognized for providing accurate absorption measurements
(Arnott et al., 1998;Lewis et al., 2008). Studies comparing photoacoustic and filter-based methods
indicate that filter-based techniques often overestimate absorption, although the AAE derived from
both methods generally aligns more closely (Al Fischer and Smith, 2018;Saleh et al., 2014).
The complex chemical composition of BrC leads to significant variability in its optical
properties. The molecular characteristics of BrC components vary based on their sources, making
specific molecular information valuable for source attribution. Studies have shown that the
molecules responsible for BrC absorption in biomass burning aerosols tend to be large and highly
unsaturated (Sun et al., 2007). Nitroaromatics, primarily including nitro-substituted benzene,
pyrrole, naphthalene, and indole derivatives, are commonly identified as BrC chromophores (Jiang
et al., 2019;Mayorga et al., 2022;Baboomian et al., 2023;Cui et al., 2024;Dalton et al., 2024),
which are either directly emitted from biomass burning or formed through atmospheric reactions
involving combustion products, nitrogen oxides, or nitrous acid (Li et al., 2014;Chen et al.,
2011;Desyaterik et al., 2013). Amines, another group of nitrogen-containing compounds, are often
detected in BrC, where they frequently serve as reactants in the formation of SOA (Nozière et al.,
2009). High-resolution mass spectrometry (HRMS) has been widely used for offline
characterization of BrC to obtain detailed molecular-level information. To improve detection
accuracy, BrC components are often separated using chromatography before MS analysis,
allowing for more precise molecular identification (Claeys et al., 2012;Zhang et al.,
2013;Desyaterik et al., 2013). To fully understand how the chemical variability of BrC influences
atmospheric radiation, it is crucial to conduct detailed chemical analyses of BrC and incorporate
the updated BrC classifications into the climate models.
Previous studies on BrC have mostly been based on laboratory simulations of typical sources,
whereas field-based measurements involving multiple BrC sources remain limited. This is partly
due to the inherent difficulty of distinguishing contributions from different sources under complex
ambient conditions, especially when using bulk sampling methods. In light of these challenges, we
adopted a case-based analysis framework to explore how different dominant source regimes and
meteorological scenarios affect the optical properties of BrC in real-world settings. In this study,
we conducted a field campaign during the summer of 2022 at Xichong site (22.48ºN, 114.56ºE),

located on the Dapeng Peninsula of Shenzhen, China. Particle optical properties and chemical compositions were measured both online and offline. Different sources of BrC were identified through meteorological data and the chemical compositions of particles. The optical properties of BrC from different sources were evaluated and compared, supported by molecular characterizations. Our study provides direct observational evidence of varying BrC sources with different optical properties in the ambient, contributing to a deeper understanding of BrC's radiative effects in climate models.

**2 METHODS**

**2.1 Field Measurements**

Field measurements were conducted at Xichong site (22.48ºN, 114.56ºE, Figure S1) on the Dapeng Peninsula in Shenzhen, China, from August to September 2022. The analysis period used in this study was from 27 August to 9 September 2022, corresponding to the overlapping operation time of all deployed instruments. Located about 60 km from the city center, Xichong site is surrounded by the sea and distant from urban areas and industrial sources, with over 90% forest coverage. Due to minimal local anthropogenic interference, Xichong site serves as a regional atmospheric background station in South China.

During the field campaign, an aethalometer (AE31, Magee Scientific, USA) operating at seven wavelengths (370, 470, 520, 590, 660, 880, and 950 nm) and a photoacoustic extinctiometer (PAX, Droplet Measurement Techniques, USA) measuring at 532 nm, were utilized to detect the online optical properties of particles. A Monitor for AeRosols and Gases in Ambient air (MARGA, Metrohm-Applikon, Netherlands) was conducted to detect the online water-soluble ion concentration ($NH_4^+$, $Na^+$, $K^+$, $Ca^{2+}$, $Mg^{2+}$, $SO_4^{2-}$, $NO_3^-$, $Cl^-$). Detailed information regarding the instrumentation and measurement uncertainties of the aethalometer is provided in the Supporting Information (SI, Text S1). In this study, the time resolution of all online data was standardized to 1 h. Offline filter sampling was also carried out simultaneously during the field campaign. A high-volume sampler (XT1025, XTrust Analytical Instruments, China) with a flowrate of $1m^3$/min was

used to collect PM$_{2.5}$ samples on the pre-baked quartz filters with sampling period of 24 h for each
filter. Details on the filter pretreatment procedures are available in the SI (Text S2). The filters
were further analyzed to measure the BrC mass, optical properties, and molecular characteristics.
Other measurements including the mass concentration of PM$_{2.5}$, O$_3$, and the meteorological
factors (temperature, relative humidity, wind speed, and wind direction), were conducted at the
sampling site. The HYSPLIT-4 (Hybrid Single-Particle Lagrangian Integrated Trajectory) model
developed by the ARL (Air Resources Laboratory) of the NOAA (National Oceanic and
Atmospheric Administration, USA) was employed to compute 24 h air mass back trajectories at a
50 m arrival height.
**2.2 Mass absorption cross-section of BrC**
The mass absorption cross-section (MAC, m$^2$/g) of BrC can be calculated according to the
following equation:
$$\text{MAC}(\lambda) = \frac{b_{\text{abs,BrC}}(\lambda)}{[\text{BrC}]} \tag{1}$$

where $b_{\text{abs,BrC}}(\lambda)$ is the light absorption coefficient (Mm$^{-1}$) of BrC at a given wavelength $\lambda$,
derived by subtracting the corresponding absorption coefficient of BC from the total particle
absorption coefficient. Here, we used both online and offline methods to calculate the MAC of
BrC.
**2.2.1 Online determination of BrC light absorption coefficient**
Previous studies have reported that the $b_{\text{abs}}$ estimated from the aethalometer is generally larger
than that measured by the PAX, likely due to artifacts associated with organic matter loading on
the filter (Lack et al., 2008;Cappa et al., 2008;Saleh et al., 2014). In this study, the correlation
between the $b_{\text{abs}}$ derived from the aethalometer ($b_{\text{abs,520}}$) and the PAX ($b_{\text{abs,532}}$) is shown in
Figure S2. The aethalometer-derived $b_{\text{abs}}$ were scaled by a factor of 2 across all wavelengths for
subsequent MAC calculations. We consider the light absorption coefficient at a wavelength of 880
nm detected by the aethalometer to be primarily attributed to BC, with minimal contribution from
BrC absorption (Laskin et al., 2015). Based on the fact that BC has minimal wavelength
dependence, with an AAE of ~1 (Bond and Bergstrom, 2006), the BC absorption coefficient at
wavelength $\lambda$, $b_{\mathrm{abs,BC}}(\lambda)$, is given by:

$$b_{\mathrm{abs,BC}}(\lambda) = b_{\mathrm{abs,BC}}(880) \times \left(\frac{\lambda}{880}\right)^{-1} \tag{2}$$

And thus the $b_{\mathrm{abs,BrC}}(\lambda)$ is calculated by:

$$b_{\mathrm{abs,BrC}}(\lambda) = b_{\mathrm{abs}}(\lambda) - b_{\mathrm{abs,BC}}(\lambda) \tag{3}$$

The light absorption coefficients of the aethalometer were not directly measured but were
converted and corrected (Text S1). In this study, we focus on wavelengths of 370 nm and 550 nm
for all the optical measurements, representing the high light absorption band and mid-visible band
of BrC, respectively, to facilitate comparisons with results from other studies. Thus, the absorption
coefficient at 520 nm wavelength detected by the aethalometer was converted to 550 nm using the
following equations:

$$b_{\mathrm{abs}}(550) = b_{\mathrm{abs}}(520) \times \left(\frac{550}{520}\right)^{-\mathrm{AAE}_{370-550}} \tag{4}$$

$$\mathrm{AAE}_{370-550} = -\frac{\ln[b_{\mathrm{abs}}(370)] - \ln[b_{\mathrm{abs}}(550)]}{\ln(370) - \ln(550)} \tag{5}$$

### 2.2.2 Offline determination of BrC mass concentration and MAC calculation

In equation (1), [BrC] is the mass concentration of BrC. Since BrC is fundamentally an optical
concept, the optical-equivalent mass of BrC can be determined according to the absorption
coefficient of BrC by assuming its MAC. Currently, there is no unified method for the direct
measurement of BrC mass. Commonly used methods for characterizing BrC include thermal
desorption and dissolution methods for characterization of BrC mass, although both come with
inherent uncertainties. The thermal desorption method quantifies BrC mass by heating the volatile
OC of the particle, taking advantage of the lower volatilization point of BrC than BC (Massabò et
al., 2016;Olson et al., 2015;Pani et al., 2021). However, it may also include some non-absorbing
OC and may induce pyrolysis during the heating process, which brings further uncertainties into
the measurement. The dissolution method measures the BrC mass after extraction in the solvent
(water, methanol, acetone, etc.) (Rathod et al., 2024). Nevertheless, some BrC may not be soluble,
which carries uncertainties to the dissolution method.
In this study, we used both thermal desorption and dissolution methods to measure the [BrC].
For the thermal desorption method, the BrC mass was measured using an organic carbon/elemental
carbon analyzer (OC/EC analyzer, DRI 2015, Magee Scientific, USA) based on the filter samples.
Detailed information on the OC/EC analyzer mechanism is provided in the SI (Text S3). The
temperature-separated carbon fractions from aerosol filter deposits were quantified for the mass
concentration of OC that evaporated up to 580°C ($[OC_T]$), which was taken as a representative of
the BrC mass concentration to calculate the MAC.
During the campaign, the optical measurement function of the OC/EC analyzer was
malfunctioning. The $b_{abs,BrC}$ values were based on online data from the aethalometer ($b_{abs,AE31}$)
with one data point per hour, whereas the $[OC_T]$ values were derived from offline filter sampling
with one data point every 24 hours. To align the temporal resolution of the data, we used $[OC_T]$
relative to the total particulate mass ($PM_{filter}$) on each filter (every 24 hours) as a fixed ratio. This
ratio was then applied to the hourly $PM_{2.5}$ mass concentration ($[PM_{2.5}]$) over the corresponding
24-hour period, yielding the calculated hourly BrC mass concentration as given by equation (3).
There might be limitations arising from the fixed BrC mass ratio ($\frac{[OC_T]}{[PM_{filter}]}$) used to calculate the
MAC over 24 hours, as the time resolution differs from the hourly $b_{abs,AE31}$. However, we believe
that the quantification of BrC mass in this study does rely on the offline filter-based analysis. The
time resolution of the online $MAC_{BrC,\lambda}$ in this study is one hour.

$$MAC_{BrC,\lambda} = \frac{b_{abs,AE31}(\lambda)}{\frac{[OC_T]}{[PM_{filter}]}\times[PM_{2.5}]} \tag{6}$$


Meanwhile, we measured the mass concentration and light absorption of water-soluble organic
carbon (WSOC). The solubility of BrC varies across different solvents. However, in this study, the
mass concentration of WSOC ($[WSOC]$) was chosen for the calculation of MAC, and BrC
dissolved in other solvents was not further discussed. The sample filters were stored at -20 °C prior
to analysis. Each filter was ultrasonically extracted in deionized water at room temperature, and
the original extract was directly used for absorbance measurements. The mass concentration of
WSOC in the collected filter samples was measured using a total organic carbon analyzer (TOC
analyzer, N/C 3100, Analytik Jena, Germany). We further compared the $[OC_T]$ detected by the
thermal desorption method and the [WSOC] measured by the dissolution method (Figure S3),
which showed good correlation ($r^2$=0.844) while the $[OC_T]$ was more than twice of the [WSOC].
The light absorption of WSOC was further measured using an ultraviolet- visible (UV-vis)
spectrometer (T2600, York Instrument, China) within the wavelength ranging from 190 to 1100
nm. The WSOC light absorption was then converted into light absorption coefficients ($b_{abs,WSOC}$),
as given by equation (4):

$$b_{abs,WSOC}(\lambda) = \ln(10) \times (A_\lambda - A_{880}) \times \frac{V_l}{V_a \times L} \tag{7}$$

where $A_{880}$ is the systematic baseline drift, $V_l$ (m$^3$) is the volume of water (30 mL) used for
extraction, $V_a$ (m$^3$) is the volume of the sampled air, and $L$ (cm) is the optical path length of the
quartz cuvette (1 cm) in the UV-vis spectrometer. The filter-based offline $MAC_{WSOC,\lambda}$ is
calculated according to:

$$MAC_{WSOC,\lambda} = \frac{b_{abs,WSOC}(\lambda)}{[wsoc]} \tag{8}$$

The MAC values reported in this study were calculated based on measured OC concentrations,
derived from both an online aethalometer and offline WSOC analysis. As these methods are
carbon-specific, using OC as the mass basis ensures consistency across the dataset. It should be
noted that some studies report MAC values normalized to organic matter (OM) rather than OC.
To convert between the two, an OM/OC ratio is typically assumed, which depends on the oxidation
state of the aerosol. Literature values suggest that OM/OC ratios range from ~1.6 to 2.5 and are
strongly correlated with the O/C ratio (Turpin and Lim, 2001;Aiken et al., 2008). Consequently,
MAC values defined per unit OC are generally higher than those defined per unit OM ($MAC_{OM}$ =
MAC$_{OC}$·[OC]/[OM]). For example, assuming an OM/OC ratio of 2.0, a MAC$_{OC}$ of 1.2 m$^2$/g would
correspond to a 0.6 m$^2$/g MAC$_{OM}$. This trend should be taken into account when comparing MAC
values across different studies.
**2.3 Chemical molecular analysis**
A high-performance liquid chromatography (HPLC) equipped with a photodiode array (PDA,
G7117C, Agilent, USA) detector and a high-resolution mass spectrometer system (HRMS,
G6545A, Agilent, USA) was utilized to identify the molecular composition, determine the relative
abundance, and measure the corresponding light absorption of BrC. The HPLC was equipped with
a C18 column (EC-C18, 3×150 mm, 2.7 μm particles, Agilent, USA), using mobile phases of 0.1%
formic acid-water (A, HPLC grade) and 0.1% formic acid-acetonitrile (B, HPLC grade). Gradient
elution for each sample was performed with the A-B mixture as: 0~1 min hold at 95% A, 1~20
min linear decreased to 5% A, 20~27 min hold at 5% A, and then 27~30 min hold at 95% A. The
HRMS was set with a soft electrospray ionization source (ESI) in full scan, operating in both
positive and negative ion modes. Raw data from mass spectrometry were processed using
MassHunter Qualitative Analysis (v10.0). Molecule concentrations were semi-quantified based on
the intensity from mass spectrometry (Kruve, 2019;Zhang et al., 2023).
The absorbance of the PDA at 370 nm was selected as the intensity of light absorption of BrC
(Hecobian et al., 2010;Wen et al., 2021). Using a peak extraction algorithm in MassHunter
Qualitative Analysis, approximately 20 absorption peaks per sample were identified. The
absorption intensity at a specific retention time was determined by subtracting the blank absorption
from the sample absorption. Peaks were grouped by overlapping retention times, with molecules
in each group recorded along with their absorbance detected by the PDA. We employed a partial
least squares regression (PLSR) model to attribute individual molecular absorbance (Text S4),
clarifying the relationship between absorbing molecules and the absorbance of individual peaks
(Zhang et al., 2023).
**3 RESULTS AND DISCUSSION**

## 3.1 Light absorption of BrC from different sources

During our sampling period, wind at the Xichong site predominantly came from two directions: northwest and northeast (Figure 1a, Figure S4). Air masses from the northwest primarily originated from inland areas of the peninsula, while those from the northeast were from the sea (Figure S1). High ozone levels were observed at times, mainly during the daytime, and were associated with relatively high nitrate concentrations (Figures 1b & 1c). Additionally, water-soluble potassium, as a marker for biomass-combusted aerosols (Zhai et al., 2015), also exhibited a time of elevated levels during our observation (Figure 1c). Based on distinct meteorological and pollutant concentration characteristics, we selected three typical cases for detailed analysis of their optical properties. The selection criteria for each case were as follows: Case 1, the ozone case, with 1) the concentration of $O_3 > 100$ ppb, 2) the concentration of $[NO_3^-] > 0.6$ μg/m$^3$, 3) wind speed $< 3$ m/s, and 4) consecutive duration $> 6$ h (red shading in Figure 1); Case 2, the transport case with 1) wind direction $>270°$, 2) wind speed $> 4$ m/s, and 3) consecutive duration $> 6$ h (blue shading in Figure 1); and Case 3, the combustion case, with 1) the concentration of $[K^+] > 0.2$ μg/m$^3$, and 2) consecutive duration $> 6$ h (yellow shading in Figure 1).

Meanwhile, the HYSPLIT 24-h air mass backward trajectories indicate that during Case 1, air masses predominantly originated from areas close to the sampling site. Combined with low wind speeds ($< 3$ m/s), meteorological conditions limited atmospheric dispersion, promoting ozone accumulation and secondary pollutant formation (Figure S5). In Case 2, the air mass trajectories were from the inland region, with strong wind speeds that facilitated the transport of pollutants to the sampling site. For Case 3, the air mass trajectories also originated from the interior region but were associated with lower wind speeds than in Case 2.

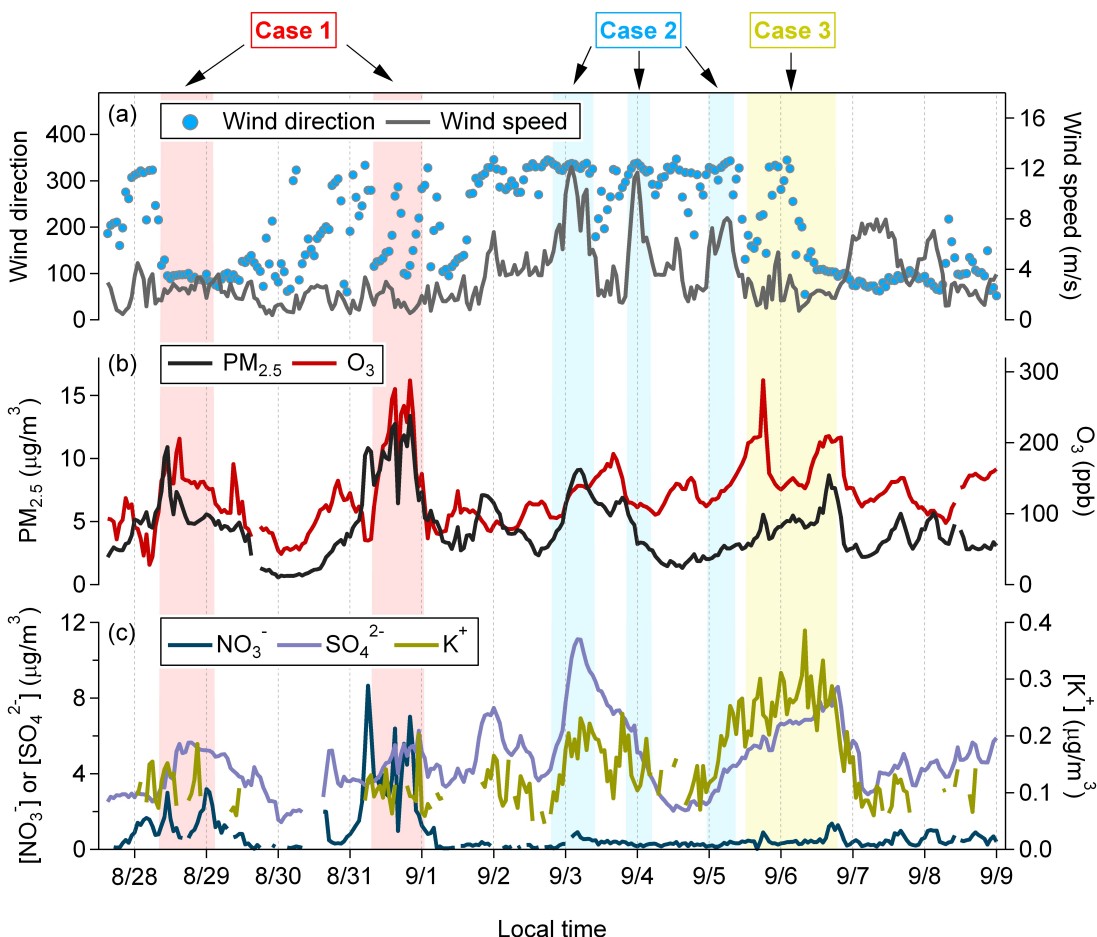

**Figure 1.** Time series of wind direction and wind speed at the sampling site (a), the concentration of $PM_{2.5}$ and $O_3$ (b), and chemical composition detected by the MARGA ($NO_3^-$, $SO_4^{2-}$, and $K^+$, d). The colored shadows denote the sampling time for the studied cases (red shading for ozone Case 1, blue shading for transport Case 2, and yellow shading for combustion Case 3).

Polar plots of wind direction, wind speed, and $MAC_{BrC,370}$ were further analyzed for Case 1–3 (Figure 2). In Case 1, the average wind speed was 2.06 m/s, with pollution mainly from local sources (Figure 2a). During Case 1, the average $PM_{2.5}$ concentration was $8.05 \pm 2.67$ µg/m³, and the average $MAC_{BrC,370}$ was $1.25 \pm 0.56$ m²/g (Figure 2d). For Case 2, the wind primarily came from the northwest, passing over the peninsula, with an average wind speed of 7.81 m/s (Figure 2b). The average $PM_{2.5}$ concentration and $MAC_{BrC,370}$ for Case 2 were $4.87 \pm 2.36$ µg/m³ and $2.68 \pm 0.30$ m²/g, respectively. In Case 3, the wind speed averaged 2.19 m/s, with erratic wind directions

(Figure 2c). The average PM$_{2.5}$ concentration and MAC$_{BrC,370}$ for Case 3 were 5.05 ± 1.32 µg/m$^3$
and 3.42 ± 0.41 m$^2$/g, respectively.
Among the three cases, although the average PM$_{2.5}$ concentration in Case 1 was the highest, its
MAC$_{BrC,370}$ was the lowest, indicating that the light-absorbing ability of BrC in this high-ozone
scenario was relatively weak. The low wind speed in Case 1 limited the influx of transported
pollutants. High concentrations of ozone and [NO$_3^-$] indicated that the aerosols in Case 1 were
primarily secondary and highly aged. However, Case 3, characterized by a high concentration of
potassium and identified as a plume from combustion sources, had the highest MAC$_{BrC,370}$ in our
observations, indicating the strongest light-absorbing ability of combusted BrC among the cases.
In real-world atmospheric environments, BrC aerosols often originate from a mixture of sources,
and complete source separation is rarely achievable in field studies. While our case-based
framework aimed to identify periods with a dominant emission influence, we acknowledge that
source mixing may still occur and introduce variability in the retrieved optical parameters. As such,
the reported MAC and AAE values should be interpreted as reflecting source-dominant conditions
rather than pure-source characteristics. Nevertheless, the clear contrasts in chemical composition
and optical responses across cases suggest that dominant sources exert a meaningful influence on
BrC absorption. This reinforces the relevance of our findings for understanding BrC behavior
under realistic ambient conditions, despite the limitations of bulk sampling and complex source
environments.

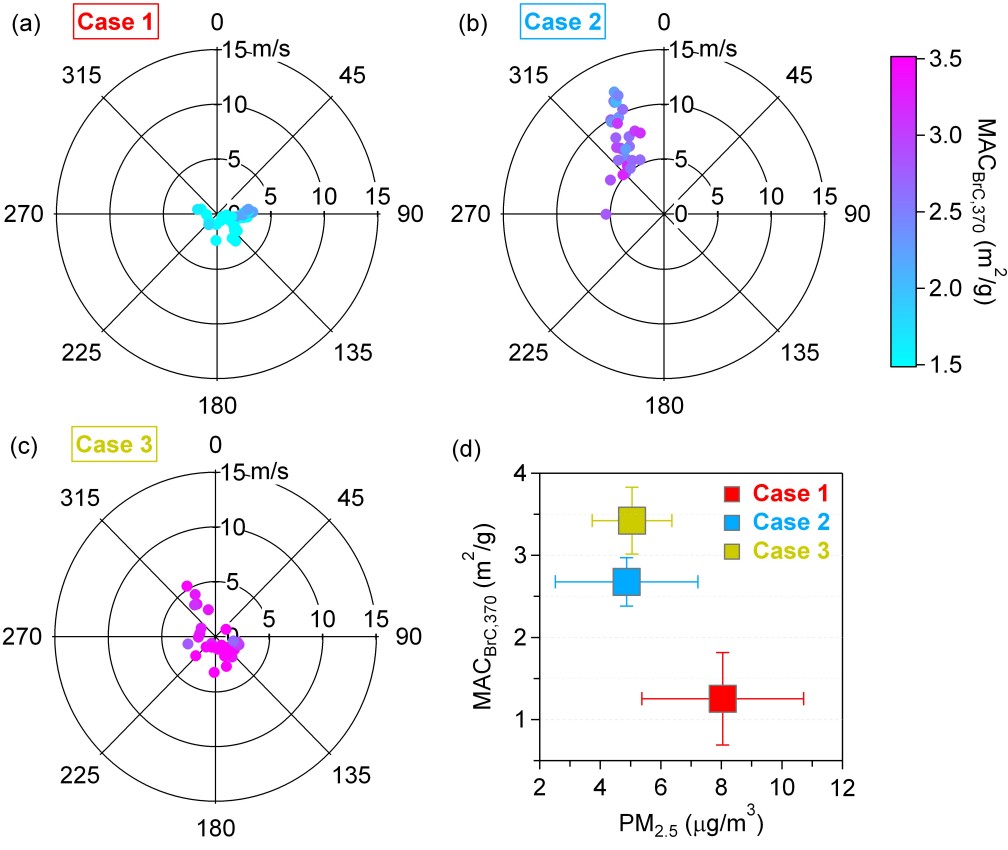

**Figure 2.** (a–c) Polar plots and $MAC_{BrC,370}$ values for Case 1–3. The radius and color represent the $MAC_{BrC,370}$ values in the downwind direction at specific wind speeds. The color scale denotes the values of $MAC_{BrC,370}$. (d) The mean $MAC_{BrC,370}$ values and mean $PM_{2.5}$ values for Case 1–3. Error bars denote a standard deviation.

### 3.2 Effects of different aerosol sources on the MAC

By compiling BrC light-absorption measurements reported in 20 studies, Saleh et al. classified BrC into four classes (Saleh, 2020), each with characteristic $MAC_{BrC,550}$ and AAE values: very weakly absorbing (VW-BrC, $MAC_{BrC,550}$ of $1.3×10^{-3}$–$1.3×10^{-2}$, AAE of 7–10), weakly absorbing (W-BrC, $MAC_{BrC,550}$ of $1.3×10^{-2}$–0.13, AAE of 5–8), moderately absorbing (M-BrC, $MAC_{BrC,550}$ of 0.13–1.3, AAE of 2.5–5), and strongly absorbing (S-BrC, $MAC_{BrC,550}$ >1.3, AAE of 1.5–2.5). The optical properties defining these BrC classes are expected to be associated with their corresponding physicochemical properties, such as molecular size, volatility, and solubility. In this

study, the AAE values for both online measurements of BrC and filter-based offline measurements
of WSOC were calculated in the wavelength range of 370 nm to 550 nm, referred to as $AAE_{370\text{-}550}$.
For online measurements of BrC, the optical results show an approximately linear correlation.
In Case 1, results fall into both the W-BrC and M-BrC categories, whereas results for Case 2 and
Case 3 fall primarily into the M-BrC category (Figure 3a). In Case 1, where the ozone
concentration is high, BrC shows weaker light-absorbing ability and stronger wavelength
dependence compared to Cases 2 & 3. BrC in Case 3 exhibits a high light-absorbing ability with
low wavelength dependence. For Case 2, a portion of the optical results overlaps with those from
Case 3, possibly due to the transported air mass originating from a similar source as in Case 3.
For the filter-based offline measurements of WSOC, the trend of $AAE_{370\text{-}550,WSOC}$ and
$MAC_{WSOC,550}$ is consistent with the online results, showing an inverse correlation (Figure 3b). The
sample in Case 1 shows the highest wavelength dependence and the lowest light-absorbing ability
of WSOC. It's worth noting that although the ozone concentration was also high during Case 3,
its optical results did not exhibit the same high wavelength dependence as observed in Case 1. The
possible reason could be that primary WSOC produced by combustion has stronger light
absorption, which dominated the optical behavior of WSOC during Case 3.
The offline MAC values based on WSOC extractions do not account for Mie scattering effects
due to the lack of particle-phase interactions in liquid measurements(Liu et al., 2013;Zeng et al.,
2020). Moreover, because particle size distribution and particle mixing state information were not
available during the sampling period, Mie model corrections were not be performed in this study.
Therefore, direct quantitative comparisons between offline and online MAC values may involve
uncertainties. Nevertheless, while the absolute MAC values from the two methods are not directly
comparable, the observed trends between the two approaches are generally consistent. This
consistency provides additional confidence in the robustness of the observed variations in BrC
optical properties across different cases.
Saleh et al. suggested that VW-BrC primarily originates from secondary BrC, W-BrC mainly
comes from smoldering BrC, and M-BrC is mainly associated with high-temperature BrC (Saleh,
2020). However, in our observations, we found that Case 1, which we consider to be dominated
by secondary BrC, still falls within the W-BrC or even M-BrC regions for both online airborne
measurements and filter-based offline analysis. Possible reasons could be: 1) Unlike laboratory
studies, field environments have greater diversity and uncertainty in BrC sources, and 2)
differences in measurement methods may lead to variations in the results (Bond and Bergstrom,
2006;Saleh, 2020). Although the results for Case 2 fall solely within the S-BrC category, we
believe that particles during this period are transported from inland urban areas, where the sources
are more complex, including contributions from traffic emissions, industrial combustion,
secondary sources, etc.
The optical properties of BrC can be affected by atmospheric aging processes such as
photochemical bleaching and secondary browning (Zhao et al., 2015). Previous studies have
demonstrated that such transformations can occur over timescales of several hours to one day,
depending on oxidant levels, radiation intensity, and humidity (Forrister et al., 2015;Washenfelder
et al., 2015). In this study, although we did not explicitly isolate aging effects, back-trajectory and
chemical evidence suggest that the BrC observed was predominantly regionally influenced, with
estimated transport times generally within this aging-relevant range. Therefore, the reported MAC
and AAE values likely represent moderately aged BrC.

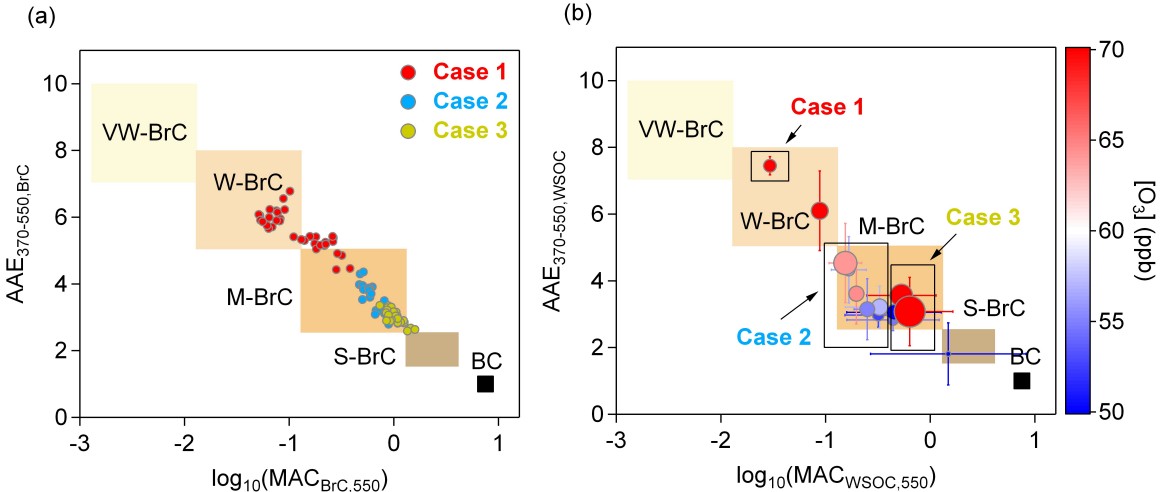

**Figure 3.** Optical-based BrC classification scheme (Saleh, 2020) in the $\log_{10}$ ($MAC_{550}$ [m$^2$/g]) vs. $AAE_{370-550}$ space for (a) BrC and (b) WSOC. The shaded areas represent very weakly absorbing BrC (VW-BrC), weakly absorbing BrC (W-BrC), moderately absorbing BrC (M-BrC), and strongly absorbing BrC (S-BrC). BC is also shown for reference (Bond and Bergstrom, 2006). The scatters in (a) correspond to the online results of Case 1–3. BrC mass concentrations used for the $MAC_{BrC,550}$ were determined based on thermally desorbed organic carbon. The scatters in (b) correspond to the filter-based results during the sampling period with each scatter representing a filter in 24 h sampling duration. The color scale in (b) denotes the ozone concentration in ppb. The size of scatters in (b) denotes the concentration of K$^+$ detected by the MARGA. Error bars denote the standard deviation of the results for three repeated experiments.

### 3.3 Chemical characterization of BrC molecules

The water-soluble organic carbon (WSOC) species were ionized using ESI+ and ESI- ionization modes to detect the organic compounds. The identified molecules were categorized into groups based on atom composition: CHO, CHON, CHOS, and CHONS. The van Krevelen (VK) diagram is a widely used graphical method that plots H/C ratios against O/C ratios in molecular formulas to qualitatively identify the major chemical species in WSOC (Kim et al., 2003). In this study, the VK space is divided into seven regions based on previous studies: (1) lipid-like (O/C = 0–0.3, H/C

= 1.5–2.0), (2) aliphatic/protein-like (O/C = 0.3–0.67, H/C = 1.5–2.2), (3) carbohydrate-like (O/C
= 0.67–1.2, H/C = 1.5–2.4), (4) unsaturated hydrocarbons (O/C = 0–0.1, H/C = 0.7–1.5), (5)
lignins/carboxylic-rich alicyclic-molecule-like (CRAM) (O/C = 0.1–0.67, H/C = 0.7–1.5), (6)
tannin-like (O/C = 0.67–1.2, H/C = 0.5–1.5), and (7) condensed aromatics (O/C = 0–0.67, H/C =
0.2–0.7) (Feng et al., 2016;Ohno et al., 2010;Zeng et al., 2024). The sizes of scatters in Figure 4
are proportional to the absorbance.
For each case, filters were selected to coincide with the core pollution periods, characterized by
stable meteorological conditions and elevated pollutant concentrations. Although the number of
samples was limited, the chemical results are considered reasonably representative of the dominant
source influences during these periods. In Case 1, CHO compounds, which account for 36.5% of
the absorbance, are the most abundant form of BrC. These CHO compounds likely contain
carboxyl or hydroxyl functional groups. The light-absorbing CHO compounds may originate from
biomass burning smoke (Desyaterik et al., 2013;Chen et al., 2022a;Zhou et al., 2022;Chen et al.,
2023) and have also been detected in water-soluble organic carbon (WSOC) and cloud water
(Bianco et al., 2018;Kourtchev et al., 2016). Secondary CHO compounds, including typical dimers
of α-pinene and diterpenoid derivatives, have also been detected in previous studies (Kourtchev et
al., 2014;Kristensen et al., 2014;Gómez-González et al., 2012). The CHOS group in Case 2
contributes the highest relative absorbance (29.5%). The CHOS compounds are considered to
contain long aliphatic carbon chains with low aromaticity and are typically derived from
anthropogenic emissions, such as diesel vehicles (Tao et al., 2014), coal combustions (Song et al.,
2019), and vessels (Cui et al., 2019). The CHON group in Case 3 exhibits the highest relative
absorbance (43.2%). It's worth noting that, although CHON is not the most abundant group in
terms of molecular abundance, its relative absorbance is the highest, suggesting that CHON
compounds have a strong molecular absorption capacity. The CHON compounds have been found
to be mainly derived from biomass burning species, such as nitrophenols, nitrocatechols,
nitroguaiacols, etc. (Kourtchev et al., 2015;Zhang et al., 2013;Song et al., 2018). Several CHON
species consistent with indole-derived structures, including isatin ($C_8H_5NO_2$) and nitroindole
($C_8H_6N_2O_2$, $C_8H_7NO_4$) were detected in the mass spectra. The identification of these compounds
supports the attribution of the observed BrC to biomass burning sources (Baboomian et al.,
2023;Chen et al., 2023;Mayorga et al., 2022;Jiang et al., 2019;Montoya-Aguilera et al., 2017), and
highlights the complexity of nitrogen-containing brown carbon species in ambient aerosols. To
provide a clearer overview of the molecular-level characteristics of BrC identified in this study,
we summarized the major light-absorbing organic compounds and structures detected in the field
samples in Table S1. We further conducted a correlation analysis between the relative absorbance
of CHON and the MAC of WSOC throughout the whole sampling period, finding that as the
relative absorbance of CHON increases, the MAC of BrC also becomes larger (Figure S7). The
measurement of chemical molecules provides support for the results corresponding to our optical
observations in different cases.

441       In the VK diagram, aliphatics, lignins, and carbohydrates dominate in all three cases. In Case 3,

WSOC shows a higher proportion of absorbance from lignins, which are commonly attributed to
biomolecules and biomass burning species (Kitanovski et al., 2014). The differences in the
chemical molecular compositions of BrC across the different cases during our observations led to
variations in the light absorption of organic matter.

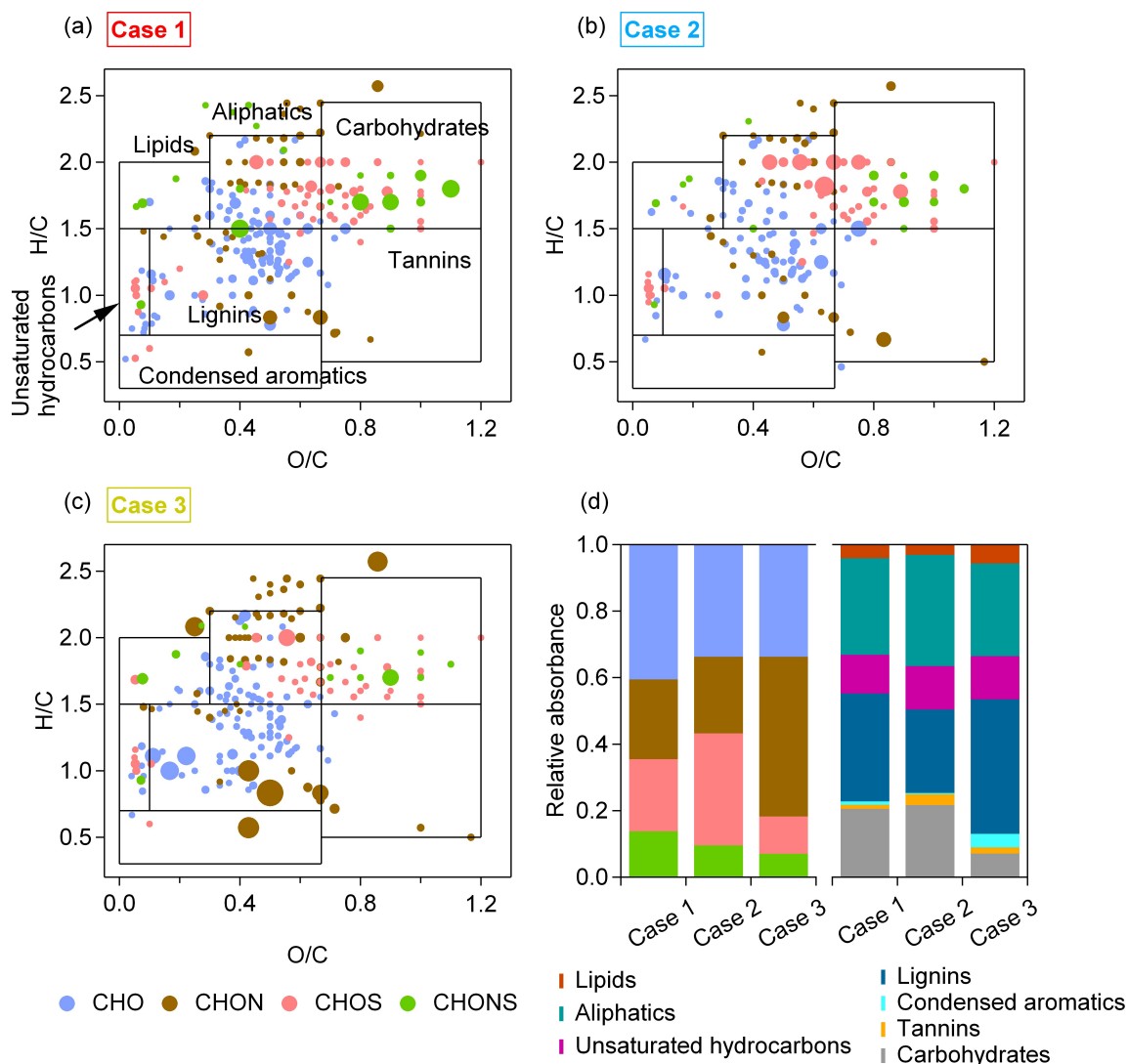

**Figure 4.** Sources of WSOC formula categories. (a–c) Van Krevelen plots for Case 1–3. Different formula categories are color coded. The sizes of scatters are proportional to the absorbance. The boxes indicate the classifications of various chemical species. (d) Relative absorbance of different formula categories (CHO, CHON, CHOS, and CHONS), and of different chemical species (lipids, aliphatics, unsaturated hydrocarbons, lignins, condensed aromatics, tannins, and carbohydrates).

## 4 CONCLUSIONS

The diverse chemical composition of atmospheric light-absorbing organics leads to distinct optical properties for BrC from different sources. However, studies on the optical properties of multi-

source atmospheric BrC, particularly those based on field observations, remain limited. The main
challenge arises from the complex and variable ambient conditions, which complicate the accurate
identification of BrC from different sources. In this study, a sampling site located away from urban
areas was selected, providing a more favorable environment for distinguishing BrC from primary
and secondary sources. Through field measurements, we independently identified various BrC
sources, including secondary BrC from ozone oxidation, primary BrC transported from urban
sources, and typical combustion-derived BrC. We found that the MAC of BrC varied by source,
with secondary BrC from ozone pollution being the least absorbing but exhibiting the highest AAE,
while BrC from biomass combustion was the most absorbing with the lowest AAE.
A key challenge in representing BrC absorption in climate models is its significant variability
in light absorption capacity. The representation of BrC absorption in climate models could be
improved by differentiating BrC sources or categorizing BrC into distinct optical ranges. Our direct
field measurements contribute to a better understanding of the optical properties of multi-source
BrC.

**Data availability.** Data used to produce the plots within this work are available in Zenodo
(https://zenodo.org/records/14780067).
**Author contributions.** JZ, XY, and PL designed the study. JZ and YZ analyzed the data. AZ and
YLZ performed the chemical molecular detections. JZ wrote the manuscript. All co-authors
contributed to discussions and suggestions in finalizing the manuscript.
**Competing interests.** The contact author has declared that none of the authors has any competing
interests.
**Acknowledgments.** The authors would like to thank the Shenzhen National Climate Observatory
for providing the observation platform for this study.
**Financial support.** This work was supported by the National Natural Science Foundation of China
(42305108, 41827804), the Guangdong Provincial Observation and Research Station for Coastal
Atmosphere and Climate of the Greater Bay Area (2021B1212050024), the Shenzhen Science and
Technology        Program        (RCBS20221008093123058,        KQTD20210811090048025,
KCXFZ20230731093601003), the Guangdong Basic and Applied Basic Research Foundation
(2025A1515011148), the Shenzhen Key Laboratory of Precision Measurement and Early Warning
Technology for Urban Environmental Health Risks (ZDSYS20220606100604008), the Opening
Project of Shanghai Key Laboratory of Atmospheric Particle Pollution and Prevention (LAP$^3$), and
High Level Special Funds (G030290001).

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
