# Peer review of "Manuscript for # Source-Dependent Optical Properties and Molecular Characteristics of Atmospheric Brown Carbon"

_EGUsphere, 2025_

## Author Response (AR1)

**Response to Reviewers**

We sincerely thank all the anonymous referees for their valuable comments and suggestions. We have extensively revised the manuscript according to the reviewers' comments. Below, we provide a point-by-point response to each comment, along with a description of the corresponding changes made in the revised manuscript.

**Anonymous Referee #1**

General comments:

The study investigates the optical properties and molecular characteristics of atmospheric brown carbon (BrC) in Shenzhen, China, focusing on how different sources impact Earth's radiation budget. BrC absorbs light in the UV-visible range, and its properties vary depending on the source. Molecular analysis revealed that biomass burning species from the CHON family exhibited the strongest absorption. These findings may help improve the understanding of BrC's radiative impact and enhance climate model accuracy. Overall, the paper is very clear and can be published after a minor revision.

**Response:** We sincerely thank Referee #1's considerate comments. In response to these suggestions, we have largely incorporated them, leading to a significant enhancement in the quality of our manuscript. Specifically, we incorporated additional discussion regarding the comparison of online and offline BrC measurements, corrected the description of nitroaromatic BrC chromophores, added technical details about filter extraction procedures, and clarified the notation for optical path length. We also addressed the differences between online and offline MAC measurements in light of Mie scattering effects and emphasized the limitations in direct comparisons. Furthermore, we confirmed the detection of key pyrrole- and indole-derived CHON species in our samples and added a new supplementary table summarizing the major BrC chromophores identified. We believe these revisions significantly strengthen the manuscript and better highlight the molecular-level insights into BrC optical properties.

Specific comments:

1. Line 77-79: For the comparison of online and offline measurements, please also refer to Chen et al. 2022a.

**Response:** We sincerely thank the reviewer for this comment. We have added further discussion regarding the comparison between online and offline measurements, as follows:

*While the varying solubility of BrC components in different solvents could introduce uncertainties in offline analyses (Shetty et al., 2019), solvent-induced chemical artifacts, particularly those associated with methanol extraction, have also been shown to significantly alter the optical properties of BrC (Kumar et al., 2018;Saleh et al., 2014). Therefore, online MAC measurements provide a more consistent and reliable benchmark, and integration with carefully selected offline extractions can offer a more comprehensive understanding of the relative abundance of BrC classes (Chen et al., 2022).*

2. Line 99-102: This sentence needs to be rewritten because of the following comments.

(1) "Nitrophenols, nitrobenzene, and their derivatives" is not an appropriate description. Nitrobenzene itself, unfortunately, only has weak absorption in the tropospheric wavelength range (wavelength above 290 nm) (Frøsig et al., 2000). Strong light absorption mainly relies on the further substitution of carbonyl or hydroxyl groups (i.e., nitrobenzaldehyde, and nitrophenol, as the author mentioned). It would be preferred to replace "Nitrophenols, nitrobenzene, and their derivatives" with "Nitroaromatics". (2) Benzene is not the only backbone for BrC molecules; pyrrole, naphthalene, and indole backbones can also be substituted by nitro groups and become nitroaromatic "BrC molecules" (Jiang et al., 2019; Mayorga et al., 2022; Baboomian et al., 2023; Cui et al., 2024; Dalton et al., 2024). Please also mention this information in the sentence. (3) Actually, "BrC molecules" is not a commonly used phrase, it would be better to replace it with "BrC chromophores".

**Response:** We appreciate the reviewer's thoughtful suggestions. The sentence has been revised accordingly, as shown below:

*Nitroaromatics, primarily including nitro-substituted benzene, pyrrole, naphthalene, and indole derivatives, are commonly identified as BrC chromophores (Jiang et al., 2019;Mayorga et al., 2022;Baboomian et al., 2023;Cui et al., 2024;Dalton et al., 2024), which are either directly emitted from biomass burning or formed through atmospheric reactions involving combustion products, nitrogen oxides, or nitrous acid (Li et al., 2014;Chen et al., 2011;Desyaterik et al., 2013).*

3. Line 216-225: Please provide some technical details about the offline preparation (e.g., filter storage temperature, filter extraction procedure).

**Response:** We thank the reviewer for the suggestion. We have added descriptions of the offline preparation process in the revised manuscript as follows:

*The sample filters were stored at -20 °C prior to analysis. Each filter was ultrasonically extracted in deionized water at room temperature, and the original extract was directly used for absorbance measurements.*

Further technical details regarding the filter pretreatment, extraction, and concentration procedures are provided in the Supplementary Information (Text S2).

***Text S2. Pretreatment of the sample filters.*** *The sample filters were punched into 1 cm$^2$ pieces for extraction. Each sample was ultrasonically extracted in deionized water at room temperature. The extract was filtered through a 0.22 μm glass fiber filter. The original extract was used directly for absorbance measurements. To achieve adequate analyte concentrations of mass spectrometry, the extraction process was repeated three times (Zhang et al., 2023). Then, the extract was freeze-dried and concentrated to 0.5 ml for further analysis.*

4. Line 222: "L (m)" should be "L (cm)" or just "L", according to the "1 cm" in Line 223.

**Response:** We thank the reviewer for identifying this error. We have corrected it by changing "L (m)" into "L (cm)" in the revised manuscript.

5. Line 314-315: Technically, the offline MAC calculated by Equation (8) may not be directly compared with the online MAC because of the Mie effect (Liu et al., 2013; Zeng et al., 2020; Zeng et al., 2021; Chen et al., 2022a). If possible, please calculate the Mie-converted offline MAC based on the reference; otherwise, if the Mie model cannot be accessed for this calculation, please provide

some descriptions in the manuscript to highlight this uncertainty.

**Response:** We thank the reviewer for highlighting this important technical issue. We fully agree that the offline MAC values derived from UV–vis measurements of filter extracts may not be directly comparable with online MAC values due to the absence of Mie scattering effects in liquid-phase measurements.

In the present study, we did not perform Mie model corrections for the offline MAC values, primarily because detailed particle size distribution data were not available for the aerosol samples corresponding to the WSOC extractions. Additionally, the internal mixing state of BrC within individual aerosol particles was also not characterized. Both the particle size distribution and mixing state information are essential inputs for accurate Mie calculations, and their absence limits our ability to perform reliable phase corrections.

In response to the reviewer's suggestion, we have added a clarification in the revised manuscript to highlight this limitation:

*The offline MAC values based on WSOC extractions do not account for Mie scattering effects due to the lack of particle-phase interactions in liquid measurements (Liu et al., 2013; Zeng et al., 2020). Moreover, because particle size distribution and particle mixing state information were not available during the sampling period, Mie model corrections were not be performed in this study. Therefore, direct quantitative comparisons between offline and online MAC values may involve uncertainties. Nevertheless, while the absolute MAC values from the two methods are not directly comparable, the observed trends between the two approaches are generally consistent. This consistency provides additional confidence in the robustness of the observed variations in BrC optical properties across different cases.*

6. Line 356-357: Please also highlight that the light-absorbing CHO compounds may come from biomass burning smoke (Desyaterik et al., 2013; Chen et al., 2022b; Zhou et al., 2022; Chen et al., 2023)

**Response:** We thank the reviewer's suggestion. We have revised the sentence as follows:

*These CHO compounds likely contain carboxyl or hydroxyl functional groups. The light-absorbing CHO compounds may originate from biomass burning smoke (Desyaterik et al., 2013;Chen et al., 2022a;Zhou et al., 2022;Chen et al., 2023) and have also been detected in water-soluble organic carbon (WSOC) and cloud water (Bianco et al., 2018;Kourtchev et al., 2016).*

7. Line 366-369: CHON from biomass burning may also involve pyrrole-derived and indole-derived species, including $C_4H_4N_2O_2$ (nitropyrroles), $C_4H_3N_3O_4$ (dinitropyrroles), $C_4H_2N_4O_6$, $C_4H_3NO_2$ (maleimide), $C_4H_3NO_3$, $C_4H_3NO_4$, $C_4H_5NO_3$, $C_4H_5NO_4$, $C_8H_5NO_2$ (isatin), $C_8H_6N_2O_2$ (nitroindole), $C_8H_7NO_4$, $C_{16}H_{10}N_2O$ (indoxyl red), and $C_{16}H_{10}N_2O_2$ (indigo dye) (Montoya-Aguilera et al., 2017; Jiang et al., 2019; Mayorga et al., 2022; Chen et al., 2022; Baboomian et al., 2023; Chen et al., 2023; Chen et al., 2024; Dalton et al., 2024; Jiang et al., 2024). Did the authors also see these compounds?

**Response:** We thank the reviewer for this valuable suggestion and for highlighting important CHON species associated with biomass burning. After re-examining our mass spectrometric data, we confirm that several pyrrole- and indole-derived compounds were detected in our samples, including

$C_8H_5NO_2$ (isatin), $C_8H_6N_2O_2$ (nitroindole), and $C_8H_7NO_4$. These compounds are consistent with previously reported markers of biomass burning emissions and secondary BrC formation.

In the revised manuscript, we have added a brief discussion acknowledging the presence of these species and their relevance to biomass burning sources:

*Several CHON species consistent with indole-derived structures, including isatin ($C_8H_5NO_2$) and nitroindole ($C_8H_6N_2O_2$, $C_8H_7NO_4$) were detected in the mass spectra. The identification of these compounds supports the attribution of the observed BrC to biomass burning sources (Baboomian et al., 2023;Chen et al., 2023;Mayorga et al., 2022;Jiang et al., 2019;Montoya-Aguilera et al., 2017), and highlights the complexity of nitrogen-containing brown carbon species in ambient aerosols.*

8. Since this paper emphasizes the "molecular characteristics" at the title, please (if possible) provide a table that summarize the major BrC molecular chromophores identified in the field samples.

**Response:** We thank the reviewer for this helpful suggestion. In response, we have added a table (Table S1) summarizing the major BrC molecular structures identified from the samples.

*The identification of molecular structures was performed by Sirius (v 5.83). Sirius operates by integrating isotope information from MS1 spectra and fragment ion information from MS2 spectra, and searching molecular structures via online databases. The absorbance spectra of the identified BrC molecules were simulated using Gaussian. Molecular geometries were optimized at the B3LYP/6-311G\*\* level to determine the most stable conformations. Subsequently, UV-Vis absorption spectra were simulated for these optimized structures using the PBE1PBE/TZVP model, with water as the solvent modeled by the IEFPCM method.*

*Molecular models were optimized and their ultraviolet-visible (UV-Vis) absorption energies were calculated using Gaussian. The resulting data were subsequently visualized with Multiwfn (v 3.8). A total of 501 theoretical single-molecule absorbance values were computed for 169 molecules. For each molecule, up to three conformers were considered, and the conformer exhibiting the highest molar absorptivity at 370 nm was selected to estimate the optical contribution. Given that the optical contribution is jointly influenced by both molar absorptivity and molecular concentration, and that the concentration of each component could not be determined in this study, a molar absorptivity threshold of 1000 was adopted to identify structures with significant absorption. Among the 169 molecules, 24 satisfied this criterion. The structural information for these molecules is summarized in Table S1.*

**Table S1.** *Molecular formulas and possible structures of the major light-absorbing contributors.*

| Formula | Mass (Da) | Absorbance | Structure |
|---------|-----------|------------|-----------|
| $C_{10}H_{10}O_4$ | 194.0579 | 69293.17 |  |

| Formula | Mass | Intensity | Structure |
|---|---|---|---|
| $C_{11}H_{10}O_6$ | 238.0477 | 45295.80 | |
| $C_{12}H_{11}N_3O_4$ | 261.0750 | 26961.74 | |
| $C_{16}H_{18}O_9$ | 354.0951 | 15911.24 | |
| $C_{16}H_{14}N_4O_5S$ | 374.0685 | 13271.89 | |
| $C_9H_8O_5$ | 196.0372 | 11023.49 | |
| $C_7H_4N_2O_3$ | 164.0222 | 10251.14 | |
| $C_8H_9NO_4$ | 183.0532 | 8124.25 | |
| $C_6H_5NO_4$ | 155.0219 | 7677.53 | |
| $C_6H_3N_3O_7$ | 228.9971 | 7596.25 | |
| $C_7H_7NO_4$ | 169.0375 | 7418.68 | |
| $C_{17}H_{20}O_8$ | 352.1158 | 6851.41 | |

| | | | |
|---|---|---|---|
| $C_7H_7NO_3$ | 153.0426 | 5679.85 |  |
| $C_8H_7NO_4$ | 181.0375 | 5214.56 |  |
| $C_7H_4N_2O_7$ | 228.0019 | 4790.78 |  |
| $C_7H_6N_2O_6$ | 214.0226 | 4587.31 |  |
| $C_7H_5NO_5$ | 183.0168 | 3414.05 |  |
| $C_7H_{10}N_6O_2$ | 210.0865 | 3385.22 |  |
| $C_6H_3N_3O_7$ | 228.9971 | 3252.82 |  |
| $C_6H_5NO_3$ | 139.0269 | 3219.28 |  |
| $C_6H_4N_2O_5$ | 184.0120 | 2604.79 |  |
| $C_9H_{10}O$ | 134.0732 | 2466.48 |  |
| $C_{16}H_{22}O_2$ | 246.1620 | 1332.55 |  |

C₁₈H₁₇N₇O₅S      443.1012      1019.21

We believe this addition provides a clearer overview of the molecular-level characteristics of BrC in our study and enhances the completeness and readability of the manuscript.

References:

Baboomian et al., Light absorption and scattering properties of indole secondary organic aerosol prepared under various oxidant and relative humidity conditions. Aerosol Sci. Technol., 2023, 57(6), 532–545. https://doi.org/10.1080/02786826.2023.2193235

Chen et al., Solvent effects on chemical composition and optical properties of extracted secondary brown carbon constituents. Aerosol Sci. Tech., 2022a, 56(10), 917–930. https://doi.org/10.1080/02786826.2022.2100734

Chen et al., Effects of Nitrate Radical Levels and Pre-Existing Particles on Secondary Brown Carbon Formation from Nighttime Oxidation of Furan, ACS Earth Space Chem. 2022b, 6, 11, 2709–2721. https://doi.org/10.1021/acsearthspacechem.2c00244

Chen et al., Contribution of Carbonyl Chromophores in Secondary Brown Carbon from Nighttime Oxidation of Unsaturated Heterocyclic Volatile Organic Compounds, Environ. Sci. Technol. 2023, 57, 48, 20085–20096. https://doi.org/10.1021/acs.est.3c08872

Chen et al., Relative Humidity Modulates the Physicochemical Processing of Secondary Brown Carbon Formation from Nighttime Oxidation of Furan and Pyrrole, ACS EST Air 2024, 1, 5, 426–437. https://doi.org/10.1021/acsestair.4c00025

Cui et al., Chemical Composition and Optical Properties of Secondary Organic Aerosol from Photooxidation of Volatile Organic Compound Mixtures, ACS EST Air, 2024, 1, 4, 247–258. https://pubs.acs.org/doi/10.1021/acsestair.3c00041

Dalton et al., Isomeric Identification of the Nitroindole Chromophore in Indole + NO3 Organic Aerosol, ACS Phys. Chem Au 2024, 4, 5, 568–574. https://pubs.acs.org/doi/10.1021/acsphyschemau.4c00044

Desyaterik et al., Speciation of "brown" carbon in cloud water impacted by agricultural biomass burning in eastern China, J. Geophys. Res. Atmos., 2013, 118, 7389–7399, doi:10.1002/jgrd.50561

Frøsig et al., Kinetics and Mechanism of the Reaction of Cl Atoms with Nitrobenzene, J. Phys. Chem. A, 2000, 104, 48, 11328–11331. https://pubs.acs.org/doi/10.1021/jp002696o

Jiang et al., Brown Carbon Formation from Nighttime Chemistry of Unsaturated Heterocyclic Volatile Organic Compounds, Environ. Sci. Technol. Lett. 2019, 6, 3, 184–190. https://pubs.acs.org/doi/10.1021/acs.estlett.9b00017

Jiang et al., Molecular analysis of secondary organic aerosol and brown carbon from the oxidation of indole, Atmos. Chem. Phys., 2024, 24, 2639–2649, https://doi.org/10.5194/acp-24-2639-2024

Liu et al., Size-resolved measurements of brown carbon in water and methanol extracts and estimates of their contribution to ambient fine-particle light absorption, Atmos. Chem. Phys., 2013, 13, 12389–12404, https://doi.org/10.5194/acp-13-12389-2013

Mayorga et al., Chemical Structure Regulates the Formation of Secondary Organic Aerosol and Brown Carbon in Nitrate Radical Oxidation of Pyrroles and Methylpyrroles, Environ. Sci. Technol., 2022, 56, 12, 7761–7770. https://pubs.acs.org/doi/10.1021/acs.est.2c02345

Montoya-Aguilera et al., Secondary organic aerosol from atmospheric photooxidation of indole, Atmos. Chem. Phys., 2017, 17, 11605–11621, https://doi.org/10.5194/acp-17-11605-2017

Zeng et al., Global Measurements of Brown Carbon and Estimated Direct Radiative Effects, Geophys. Res. Lett., 2020, 47, e2020GL088747, https://doi.org/10.1029/2020gl088747

Zeng et al., Assessment of online water-soluble brown carbon measuring systems for aircraft sampling, Atmos. Meas. Tech., 2021, 14, 6357–6378, https://doi.org/10.5194/amt-14-6357-2021

Zhou et al., Molecular Characterization of Water-Soluble Brown Carbon Chromophores in Snowpack from Northern Xinjiang, China, Environ. Sci. Technol. 2022, 56, 7, 4173–4186. https://doi.org/10.1021/acs.est.1c07972

**Anonymous Referee #2**

"Source-Dependent Optical Properties and Molecular Characteristics of Atmospheric Brown Carbon" describes measurements of the composition and optical properties of BrC aerosols sampled during the summer in Shenzhen, China. Different sources of BrC are proposed based on the measurements that have different mass absorption cross-sections (MAC). BrC is a complex topic that is of importance due to its role in understanding the radiative budget. Understanding real-world sources of BrC and constraining its optical properties is an area of active investigation that would greatly benefit from more data. While the underlying data from this manuscript may be able to contribute to and advance our understanding of BrC, it is my opinion that the analysis is insufficiently developed. Substantial additional analysis is required and, in my opinion, publication at this time is premature. Below I outline the main reasons.

**Response:** We sincerely thank the reviewer for the thorough and constructive feedback. In response to the major and minor comments, we have carefully revised the manuscript to address the reviewer's concerns. Specifically, we clarified the case selection criteria, added discussions on source mixing and BrC aging effects, acknowledged the limitations related to offline and online MAC comparisons, and explained the choice of OC-based normalization for MAC calculations. We also included additional chemical evidence (e.g., detection of indole-derived CHON species) to strengthen the source attribution for Case 3 and added a new table summarizing key BrC chromophores identified in the samples. Moreover, we revised the manuscript to accurately reflect the measurement period and to highlight both the scope and limitations of this initial case-based study. We believe these revisions significantly improve the clarity, rigor, and contribution of the manuscript.

Main comments

In my opinion, the main contribution that this manuscript attempts to make is in providing source specific MAC and AAE for three different BrC sources. However, the attribution of the measurements to these specific sources in tenuous and requires substantial further justification. The sources are identified based on classification of 3 cases. From what I can tell, the classification was made based on only a few parameters (for Case 1, ozone, particulate nitrates, wind speed, and duration; for Case 2; wind direction, wind speed, duration; Case 3; concentration of particulate potassium, duration). This limited number of variables is insufficient for source attribution, particularly in a complex environment and if the goal is to provide profile information that is representative of a given source. There may be sufficient data already collected (i.e. the mass spectral measurements) to do a more accurate source apportionment.

**Response:** We appreciate the reviewer's suggestion to use mass spectral data to perform a more rigorous source apportionment, followed by analysis of BrC properties based on identified source profiles. This is indeed a valuable approach used in field atmospheric chemistry studies. However, such source apportionment methods typically require either (1) high time-resolution online mass spectrometry data (e.g., from the AMS) or (2) a long-term offline dataset (e.g., multi-month filter-based sampling) to ensure a robust statistical base for factor analysis (e.g., PMF). In the original manuscript, our primary aim was to conduct a case-based exploratory analysis rather than a strict source apportionment. The classification relied on key tracers and meteorological conditions that are commonly used as qualitative indicators for identifying dominant source influences.

Moreover, unlike controlled laboratory experiments where source inputs and environmental conditions are tightly regulated, field studies usually face complex and variable atmospheric conditions. Even with mass spectral apportionment, it is often difficult to isolate signals from a single dominant source due to the bulk nature of the measurements (rather than single-particle resolution).

Instead, our study adopts a case-based approach, in which representative pollution episodes were identified based on meteorological parameters (e.g., wind direction, speed, pollutant levels). These cases were then compared in terms of their BrC optical properties. Our subsequent chemical characterization using offline mass spectrometry confirmed the dominant source for each case. While we do not claim to provide quantitative source apportionment, we believe our approach still offers meaningful insights into how different atmospheric conditions and dominant sources influence the optical properties of BrC in a real-world atmospheric environment. To make it clearer, we have added a discussion as follows:

*Previous studies on BrC have mostly been based on laboratory simulations of typical sources, whereas field-based measurements involving multiple BrC sources remain limited. This is partly due to the inherent difficulty of distinguishing contributions from different sources under complex ambient conditions, especially when using bulk sampling methods. In light of these challenges, we adopted a case-based analysis framework to explore how different dominant source regimes and meteorological scenarios affect the optical properties of BrC in real-world settings.*

In real-world samples, multiple BrC sources are likely to be mixed together. This fact and how it affects the results is insufficiently described. I am not convinced by the data presented that each period could be attributed to a single BrC source. How mixing of various sources affects the retrieved BrC properties is essential for accurately reporting profiles. Without consideration of the mixing of BrC sources, the results (in their current state) are of limited use to the community.

**Response:** We thank the reviewer for this important comment. We fully agree that in real-world atmospheric samples, multiple BrC sources are often mixed, and that such mixing can influence the retrieved optical properties.

In the original manuscript, the case selections were based on dominant tracer species and meteorological conditions indicative of specific sources. However, we acknowledge that complete isolation of a single source is unlikely in a complex urban environment. To address this concern, we have added a discussion to explicitly consider the potential effects of source mixing:

*In real-world atmospheric environments, BrC aerosols often originate from a mixture of sources, and complete source separation is rarely achievable in field studies. While our case-based framework aimed to identify periods with a dominant emission influence, we acknowledge that source mixing may still occur and introduce variability in the retrieved optical parameters. As such, the reported MAC and AAE values should be interpreted as reflecting source-dominant conditions rather than pure-source characteristics. Nevertheless, the clear contrasts in chemical composition and optical responses across cases suggest that dominant sources exert a meaningful influence on BrC absorption. This reinforces the relevance of our findings for understanding BrC behavior under realistic ambient conditions, despite the limitations of bulk sampling and complex source environments.*

To improve the understanding of BrC sources, it is important to consider how representative the results of a case study such as this are. Knowing the variability of BrC characteristics for a given source is critical for achieving the outcomes hat the manuscript identifies as the driving science questions (e.g., improving models, etc.). The failure to show the variability could be impacted by the limited duration of the measurement, but the broad averaging approach taken for assigning BrC sources (rather than more specific source apportionment) also contributes. It may be possible to understand more of the variability in the data set and thus improve the impact of the work by applying statistical analysis of the data and considering more of the days.

**Response:** We thank the reviewer for raising this issue. We agree that understanding the variability within a given source category is crucial. In this study, our primary objective was not to produce a comprehensive statistical representation of each BrC source, but rather to explore the differences in BrC optical properties under distinct, source-dominated pollution scenarios. We selected representative cases to reflect contrasting dominant sources and meteorological conditions, which enabled us to examine source-related differences in MAC and AAE values.

In field conditions, especially under complex and dynamic meteorological regimes, a stable dominant source rarely persists over long durations, typically lasting only a few hours, though in some exceptional cases (e.g., dust storms or large-scale wildfires), it may extend over tens of hours. While our filter sampling was conducted on a daily basis, the optical properties presented in this study were derived from high-resolution online measurements (hourly resolution). This finer temporal resolution, combined with coherent meteorological conditions and supporting chemical indicators during the selected episodes, strengthens our confidence that the BrC optical properties reported for each case are representative of source-dominant periods.

The manuscript insufficiently discusses how aging (browning or bleaching) would affect the results. What are the timescales of aging are anticipated? How do those compare to what is known about rates of browning/bleaching?

**Response:** We thank the reviewer for the valuable comment regarding the potential impact of BrC aging processes, including browning and bleaching, on the observed optical properties. We agree that BrC can undergo significant changes in its optical characteristics (e.g., MAC and AAE) due to photochemical and multiphase aging in the atmosphere, which is highly relevant for interpreting field observations.

In laboratory studies, such aging effects can be systematically investigated under controlled conditions, allowing to isolate the influence of browning or bleaching on BrC absorption. However, in field environments, the highly variable meteorological conditions, complex precursor mixtures, and overlapping reaction pathways make it challenging to quantitatively separate the effects of aging from those of primary emissions. In certain extreme events, such as wildfires, where emission plumes are relatively concentrated and background interference is minimal, the evolution of BrC optical properties during atmospheric processing can be more clearly observed (e.g., Forrister et al., 2015; Washenfelder et al., 2015). In contrast, the urban and regional setting of our study features diverse BrC sources and complex meteorological conditions, where primary emissions and aging processes often coexist.

In response to the reviewer's suggestion, we have added some discussions as follows:

*The optical properties of BrC can be affected by atmospheric aging processes such as photochemical bleaching and secondary browning (Zhao et al., 2015). Previous studies have demonstrated that such transformations can occur over timescales of several hours to one day, depending on oxidant levels, radiation intensity, and humidity (Forrister et al., 2015;Washenfelder et al., 2015). In this study, although we did not explicitly isolate aging effects, back-trajectory and chemical evidence suggest that the BrC observed was predominantly regionally influenced, with estimated transport times generally within this aging-relevant range. Therefore, the reported MAC and AAE values likely represent moderately aged BrC.*

It appears that MAC values are reported based on organic carbon mass rather that organic aerosol mass. The implications of that difference and how it affects comparison between these results and results from the literature should be clearly stated to avoid the misuse of the results by future studies.

**Response:** We appreciate the reviewer's comment regarding the mass basis used for MAC calculations. In this study, MAC values were calculated using organic carbon (OC) mass, rather than organic aerosol (OA) mass. This choice was made to maintain consistency with our analytical methods: both the online aethalometer and the offline WSOC analysis are based on carbon-specific quantification. As such, reporting MAC per unit OC is the most accurate and internally consistent approach for our dataset.

We agree that this distinction may affect comparisons with other studies that report MAC based on OA mass. To address this, we have added clarification in the revised manuscript:

*The MAC values reported in this study were calculated based on measured OC concentrations, derived from both an online aethalometer and offline WSOC analysis. As these methods are carbon-specific, using OC as the mass basis ensures consistency across the dataset.*

Minor comment

The manuscript does not clearly communicate the length of the measurement period (~2 weeks judging by Fig. 1). The wording in the abstract (line 27 "summer"), introduction (line 114 "summer"), and methods (line 125 "August to September") all imply that the measurement period is much longer than it was. These statements should be updated to not mislead the reader.

**Response:** We appreciate the reviewer's comment regarding the clarity of the measurement period. The actual analysis period used in this study was from 27 August to 9 September 2022, lasting approximately two weeks. This period was selected based on the overlapping operational time of all instruments used in the study, in order to ensure data consistency across optical and chemical measurements.

We have now revised the relevant descriptions in the manuscript:

*The analysis period used in this study was from 27 August to 9 September 2022, corresponding to the overlapping operation time of all deployed instruments.*

Summary: The underlying data presented seems potentially useful. However, further analysis is required for the results to be of use/application beyond this one study. Given the data collected, I think a more nuanced analysis of the measurements is potentially possible and such results could be potentially publishable.

**Response:** We thank the reviewer for recognizing the potential value of the dataset presented in this study. We agree that further analysis could yield additional insights and expand the applicability of the results. However, the current analysis was intentionally designed as an initial investigation to explore the optical properties of BrC under distinct, source-dominated ambient conditions using a case-based approach.

In response to the reviewer's suggestion, we have strengthened the discussion section to clarify the scope and limitations of the present study, and to more explicitly highlight the potential of the dataset for future in-depth investigations (e.g., source apportionment, aging analysis). While we acknowledge that more complex analysis, such as multivariate statistical modeling or longer-term sampling, may enhance the generalizability of the conclusions, we believe the current results still provide meaningful field-based insights into BrC optical behavior under representative real-world scenarios.

**Anonymous Referee #3**

The atmospheric conditions are complex and changeable, considering both the emissions and meteorological conditions. So it could be considered "super lucky" to capture three different feature events in field measurements which is only 12 days long. However, based on what the authors currently show in the paper, the determination of different events, at least Case 3, is vague. I think this study could contribute to the research on BrC, if the authors could address the following comments properly.

**Response:** We sincerely thank Referee #3 for the thoughtful and constructive comments. In response, we have carefully revised the manuscript to address the concerns raised. Specifically, we reorganized Section 2.2 into two sub-sections to clearly separate the online and offline BrC measurement methods, corrected the axis label in Figure 1c for clarity, and strengthened the source characterization for Case 3 by confirming the presence of indole-derived CHON species in the mass spectra. We believe these revisions enhance the structure, clarity, and scientific rigor of the manuscript.

Section 2.2 is too long. It is suggested that the section 2.2 could be split into two parts, i.e., online and offline determination of BrC, respectively, to make it easier to read.

**Response:** We thank the reviewer for this helpful suggestion. In the revised manuscript, we have restructured Section 2.2 by dividing it into two sub-sections:

*Section 2.2.1: Online determination of BrC light absorption coefficients and MAC*

*Section 2.2.2: Offline determination of BrC mass concentration and MAC calculation*

This revised structure provides a clearer separation between the online and offline analytical approaches and improves the overall readability of the methods section. We hope this modification enhances clarity and makes it easier for readers to follow the procedures used in this study.

Figure 1: the axis label of left axis in Figure 1c is suggested to revise. The current expression may lead to some understanding, e.g., the ratio of nitrate/sulfate.

**Response:** Thank you for pointing out the ambiguity in the y-axis label of Figure 1c. To avoid confusion with a ratio expression, we have revised the label by replacing the "/" with "or", making the meaning clearer to the reader. The updated figure is included in the revised manuscript.

[Figure]

Lines 250-263: K+ may also come from sources other than biomass burning, e.g., fireworks, so the determination of Case 3 should be with caution. Since the authors have collected filters, the authors are suggested to determine the abundances of other biomass burning tracers to help support the identification of Case 3.

**Response:** We thank the reviewer for highlighting this important point. We have re-examined our mass spectrometric data and confirmed the detection of several indole-derived CHON species, including $C_8H_5NO_2$ (isatin), $C_8H_6N_2O_2$ (nitroindole), and $C_8H_7NO_4$. These compounds are consistent with previously reported markers of biomass burning emissions and secondary BrC formation (Baboomian et al., 2023; Chen et al., 2023; Mayorga et al., 2022; Jiang et al., 2019; Montoya-Aguilera et al., 2017).

The identification of these nitrogen-containing chromophores supports the attribution of Case 3 to biomass burning influences and highlights the chemical complexity of ambient BrC species. In addition, the classification of Case 3 was based on combined evidence from elevated $K^+$ concentrations, increased $PM_{2.5}$ levels, stagnant meteorological conditions, and enhanced BrC light absorption features. We have revised the manuscript and added a brief discussion acknowledging the presence of CHON species and their relevance to biomass burning sources:

*Several CHON species consistent with indole-derived structures, including isatin ($C_8H_5NO_2$) and nitroindole ($C_8H_6N_2O_2$, $C_8H_7NO_4$) were detected in the mass spectra. The identification of these compounds supports the attribution of the observed BrC to biomass burning sources (Baboomian et al., 2023;Chen et al., 2023;Mayorga et al., 2022;Jiang et al., 2019;Montoya-Aguilera et al., 2017), and highlights the complexity of nitrogen-containing brown carbon species in ambient aerosols.*

Section 3.1: since the title is "light absorption of BrC", it would be more appropriate to add some discussions on results from water-soluble absorption as well.

**Response:** We thank the reviewer for this valuable suggestion. In this study, the light absorption properties of WSOC (offline measurements) were not directly compared with those from online measurements because the WSOC absorption results do not account for Mie scattering effects inherent to particle-phase measurements. Furthermore, due to the lack of particle size distribution data, Mie model corrections could not be performed. In addition, the offline WSOC analysis had a lower time resolution (24-hour integration) compared to the hourly online absorption measurements, limiting its ability to capture short-term variations. Therefore, the offline WSOC absorption results were used as a supplementary verification rather than as a direct comparison, and are discussed separately in Section 3.2.

To address this important point, we have added a clarification in the revised manuscript, explicitly discussing the differences between online and offline measurements and highlighting the uncertainties associated with their comparison:

*The offline MAC values based on WSOC extractions do not account for Mie scattering effects due to the lack of particle-phase interactions in liquid measurements (Liu et al., 2013;Zeng et al., 2020). Moreover, because particle size distribution data were not available during the sampling period, Mie model corrections could not be performed in this study. Therefore, direct quantitative comparison between offline and online MAC values may introduce uncertainties. Nevertheless, we note that although absolute MAC values from online and offline methods are not directly comparable, the observed trends between the two approaches are generally consistent. This consistency provides additional confidence in the robustness of the observed variations in BrC optical properties across different cases.*

Figure 3: it is suggested to use different expressions of AAE370-550 for online and offline results.

**Response:** Thank you for the helpful suggestion. To clearly distinguish between the $AAE_{370-550}$ values derived from online and offline measurements, we have updated the labels in Figure 3. Specifically, we now use "BrC" to represent the online AAE results based on aethalometer data, and "WSOC" to denote the offline AAE results based on UV-vis measurements of filter extracts.

[Figure]

For the chemical characterization, as the collection period for each filter is 24 hours, does it mean that the compounds determined were only from one or two samples for each Case? Then is it representative for this Case?

**Response:** We thank the reviewer for raising this important point. Indeed, due to the 24-hour collection period for each filter, only one or two filter samples corresponded to each identified case. We acknowledge that this limits the temporal resolution of the chemical characterization.

To minimize this uncertainty, we carefully selected filters that coincided with the peak or most stable periods of each pollution episode, based on meteorological parameters and pollutant concentration trends. Thus, although the sample number is limited, we believe the selected filters are reasonably representative of the dominant conditions for each case.

To further address the potential uncertainty, we have added clarification and discussion of this limitation in the revised manuscript:

*For each case, filters were selected to coincide with the core pollution periods, characterized by stable meteorological conditions and elevated pollutant concentrations. Although the number of samples was limited, the chemical results are considered reasonably representative of the dominant source influences during these periods.*

---

## Author Response (AR2)

**Response to Editor**

We sincerely thank the editor for the valuable suggestions and comments. We have carefully revised the manuscript to address the points raised. Detailed point-by-point responses to each of the editor's comments are provided below.

A notation of the MAC values calculated based on the organic carbon mass needs to be clearly stated in the abstract.

**Response:** We thank the editor for this insightful comment. In the revised manuscript, we have clarified the basis of MAC calculation in the abstract by adding the following sentence:

*BrC mass concentrations were determined either based on thermally desorbed organic carbon or water-soluble organic carbon, and the corresponding mass absorption cross-sections (MAC) were calculated accordingly.*

We believe this revision ensures that readers are aware of the basis for MAC determination from the outset.

Lines 162-163: method for the 'subtracting the absorption coefficient attributed to BC from the total particle absorption coefficient' needs to be elaborated and included either in the paper itself or in the SI file.

**Response:** We thank the editor for this comment. In fact, the method for subtracting the absorption coefficient attributed to BC from the total particle absorption coefficient was already described in our original manuscript (Line 171-178), as shown below:

*We consider the light absorption coefficient at a wavelength of 880 nm detected by the aethalometer to be primarily attributed to BC, with minimal contribution from BrC absorption (Laskin et al., 2015). Based on the fact that BC has minimal wavelength dependence, with an AAE of ~1 (Bond and Bergstrom, 2006), the BC absorption coefficient at wavelength $\lambda$, $b_{abs,BC}(\lambda)$, is given by:*

$$b_{abs,BC}(\lambda) = b_{abs,BC}(880) \times (\frac{\lambda}{880})^{-1} \qquad (2)$$

*And thus the $b_{abs,BrC}(\lambda)$ is calculated by:*

$$b_{abs,BrC}(\lambda) = b_{abs}(\lambda) - b_{abs,BC}(\lambda) \qquad (3)$$

To avoid disrupting the structure of the manuscript, we have retained this content in Section 2.2.1, and we hope this addresses the editor's concern.

Lines 239-241: A relationship between MAC defined based on the organic carbon (OC) versus organic mass (OM) needs additional elaboration, discussing plausible ratios between MAC(OC) and MAC(OM) based on the literature reports and characteristic O/C values allowing to estimate OM from OC. A general trend of MAC(OC)>MAC(OM) needs to be noted.

**Response:** We thank the editor for this valuable comment. We have added a detailed explanation in the revised manuscript to clarify the relationship between MAC values defined on an OC basis versus an OM basis:

*It should be noted that some studies report MAC values normalized to organic matter (OM) rather than OC. To convert between the two, an OM/OC ratio is typically assumed, which depends on the oxidation state of the aerosol. Literature values suggest that OM/OC ratios range from ~1.6 to 2.5 and are strongly correlated with the O/C ratio (Turpin and Lim, 2001;Aiken et al., 2008). Consequently, MAC values defined per unit OC are generally higher than those defined per unit OM ($MAC_{OM} = MAC_{OC} \cdot [OC]/[OM]$). For example, assuming an OM/OC ratio of 2.0, a $MAC_{OC}$ of 1.2 $m^2/g$ would correspond to a 0.6 $m^2/g$ $MAC_{OM}$. This trend should be taken into account when comparing MAC values across different studies.*

Figure 3: A note of the OC-defined MAC and units of MAC values need to be included in the figure caption.

**Response:** We thank the editor for this suggestion. We have revised the caption of Figure 3 to (1) indicate that the MAC values were calculated based on thermally desorbed organic carbon, and (2) clarify the units by presenting $MAC_{550}$ in the form of $\log_{10}(MAC_{550}$ [$m^2/g$]):

*__Figure 3.__ Optical-based BrC classification scheme (Saleh, 2020) in the $\log_{10}$ ($MAC_{550}$ [$m^2/g$]) vs. $AAE_{370-550}$ space for (a) BrC and (b) WSOC. The shaded areas represent very weakly absorbing BrC (VW-BrC), weakly absorbing BrC (W-BrC), moderately absorbing BrC (M-BrC), and strongly absorbing BrC (S-BrC). BC is also shown for reference (Bond and Bergstrom, 2006). The scatters in (a) correspond to the online results of Case 1–3. BrC mass concentrations used for the $MAC_{BrC,550}$ were determined based on thermally desorbed organic carbon. The scatters in (b) correspond to the filter-based results during the sampling period with each scatter representing a filter in 24 h sampling duration. The color scale in (b) denotes the ozone concentration in ppb. The size of scatters in (b) denotes the concentration of $K^+$ detected by the MARGA. Error bars denote the standard deviation of the results for three repeated experiments.*

Figure S2: units of absorbance coefficients need to be added to either caption or legends.

**Response:** We have added the unit $Mm^{-1}$ for the absorption coefficients in the caption of Figure S2 to improve clarity.

*__Figure S2.__ The correlation of absorption coefficients ($Mm^{-1}$) derived from the aethalometer ($b_{abs,520}$) and the PAX ($b_{abs,532}$).*

Figure S3: units of mass concentrations need to be added to either caption or legends.

**Response:** Thank you for the suggestion. We have added the unit $\mu g/m^3$ for mass concentrations in the caption of Figure S3.

*__Figure S3.__ The correlation of BrC mass concentration ($\mu g/m^3$) detected by the thermal desorption method ($[OC_T]$) and the dissolution method ($[WSOC]$).*

Figures S6-7: A note of the OC-defined MAC and units of MAC values need to be included in the figure caption.

**Response:** Thank you for the helpful comment. We have updated the captions of Figures S6 and S7 to indicate that the MAC values were calculated based on either thermally desorbed organic carbon or water-soluble organic carbon, and we have added the corresponding units as [$m^2/g$].

***Figure S6.*** *Optical-based BrC classification scheme (Saleh, 2020) in the log10 ($MAC_{BrC,550}$ [$m^2$/g]) vs. $AAE_{370-550}$ space for online measurements throughout the whole sampling period. BrC mass concentrations used for the $MAC_{BrC,550}$ were determined based on thermally desorbed organic carbon. The color scale denotes the concentration of ozone in ppb. The size of scatters denotes the concentration of $K^+$ detected by the MARGA.*

***Figure S7.*** *Relative absorbance of CHON detected in WSOC vs. log10 ($MAC_{WSOC,550}$ [$m^2$/g]) from offline filter-based measurements throughout the whole sampling period. BrC mass concentrations used for the $MAC_{WSOC,550}$ were determined based on water-soluble organic carbon. The color scale denotes the concentration of ozone in ppb. The size of scatters denotes the concentration of $K^+$ detected by the MARGA.*

Table S1: meaning of the listed absorbance values is unclear. Relevant descriptions and units need to be included.

**Response:** We thank the editor for pointing this out. In Table S1, the listed "absorbance" values refer to the simulated molar absorption intensities (i.e., molar absorptivity, ε, in L·mol$^{-1}$·cm$^{-1}$) at the electronic transition wavelengths calculated using TD-DFT (time-dependent density functional theory) in Gaussian 16. These values were extracted and visualized using Multiwfn (v3.8), and reflect the intrinsic light-absorbing capacity of each molecule under isolated conditions. We have added a note in the caption of Table S1:

***Table S1.*** *Molecular formula, molecular mass (Da), simulated molar absorptivity (L·mol$^{-1}$·cm$^{-1}$) at 370 nm, and proposed structures of major light-absorbing BrC chromophores identified in this study.*